# Guided Navigation in Knowledge-Dense Environments: Structured Semantic Exploration with Guidance Graphs

## Abstract

While Large Language Models (LLMs) exhibit strong linguistic capabilities, their reliance on static knowledge and opaque reasoning processes limits their performance in knowledge-intensive tasks. Although Knowledge Graphs (KGs) can mitigate this, existing retrieval methods are caught in a fundamental granularity trap: query-guided search leads to wasteful redundancy, while clue-guided traversal struggles with contextual reasoning in multi-hop scenarios. To address these limitations, we propose Guidance-Graph-guided Knowledge Exploration (GG-Explore), a novel framework that introduces an intermediate Guidance Graph to bridge unstructured queries and structured knowledge retrieval. This Guidance Graph acts as a lightweight semantic blueprint, abstracting the structure of potential answers to constrain the search space without sacrificing contextual breadth. Leveraging this graph, GG-Explore employs a hybrid pruning strategy, combining a model-free Structural Alignment that filters candidates using graph constraints with a Context-Aware Semantic Alignment module that refines the results by enforcing semantic consistency. Extensive experiments show our method achieves superior efficiency and outperforms SOTA, especially on complex tasks, while maintaining strong performance with smaller LLMs, demonstrating practical value.

## 1 Introduction

Large Language Models (LLMs) have demonstrated remarkable capabilities across diverse natural language tasks, including question answering Wang et al. (2024b); Li et al. (2024); Zhao et al. (2024), text generation Ji et al. (2024); Chen et al. (2023; 2022); Gong & Sun (2024), and recommender systems Zhang et al. (2023; 2024); Wu et al. (2024); Wang et al. (2024c). By leveraging deep learning and vast training data, they achieve human-like language fluency. However, their knowledge is static post-training, limiting real-time updates; they may generate plausible but incorrect responses ("hallucinations") Bang et al. (2023); Ji et al. (2023); Luo et al. (2023b). A promising solution is to ground LLMs in external, structured knowledge sources, such as Knowledge Graphs (KGs)Zhang et al. (2019b); Yao et al. (2019); Wang et al. (2021); Luo et al. (2023a). However, effectively retrieving and integrating knowledge from KGs remains a significant challenge

Some researchers have attempted to directly employ LLMs to transform questions into database query statements for retrieving relevant knowledge from knowledge graphs Hu et al. (2023); Wang et al. (2023). While this approach is straightforward, the generated queries may not always align well with the structure of the knowledge graph. A viable alternative provides KG context to the LLM, using question-guided multi-round iteration to progressively identify relevant knowledge Li et al. (2023d); Zhao et al. (2023); Sun et al. (2024). Although more robust, this approach is plagued by a fundamental granularity mismatch: the natural language question provides broad semantic intent, while KG retrieval requires precise entity and relation identifiers. This mismatch inevitably causes imprecise knowledge targeting and computational inefficiency due to redundant and sprawling graph traversals.

FiSKE have attempted to address the efficiency issue by extracting fine-grained keywords from the query to constrain the search space, thereby reducing redundancy Tao et al. (2025). However, this

strategy introduces a new critical weakness: by discarding broader contextual information for the sake of precision, it becomes myopic to the intricate relational structures essential for answering complex questions, as illustrated in the Figure 1. This illustrates a key insight: effective exploration requires not just precision, but also awareness of the broader semantic and structural context.

To overcome these limitations, we propose GG-Explore, a novel framework that introduces a Guidance Graph as an intermediate representation to bridge the semantic gap between unstructured queries and structured KGs. Instead of directly searching the massive KG, GG-Explore first constructs a lightweight, query-specific Guidance Graph. This graph abstracts the crucial relational structure of the anticipated answer while preserving the wider semantic context of the original question. It thus acts as a semantic blueprint that precisely defines the retrieval space. Unlike methods that prune based on isolated entities/relations, guided by this blueprint, our framework employs two efficient mechanisms: (1) Structural Alignment, which performs a rapid, model-free filtering of KG subgraphs —both before and after semantic checks— to eliminate structurally incompatible candidates; and (2) Context-Aware Semantic Alignment, which refines the results by enforcing semantic consistency. This integrated process ensures that retrieval is both highly precise and contextually grounded.

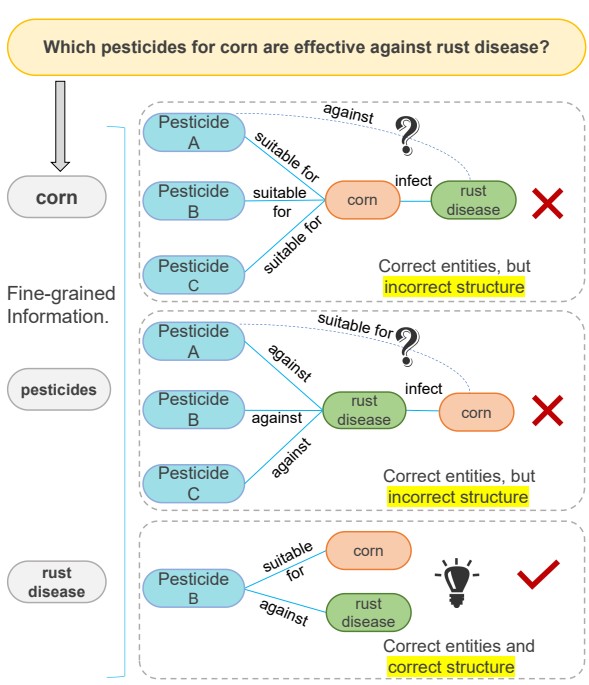

Figure 1: Relying solely on fine-grained information may fail to differentiate paths containing identical entities but distinct relational structures.

The contributions of this paper are as follows:

1. We propose the Guidance Graph, a novel intermediate representation that serves as a semantic blueprint to resolve this mismatch by abstracting structural constraints while preserving contextual semantics.

2. We develop two efficient, synergistic mechanisms—Structural Alignment and Context-Aware Semantic Alignment—that leverage the Guidance Graph to achieve precise and efficient knowledge retrieval.

3. Extensive experiments demonstrate the effectiveness of our method, with three key advantages: (1) significantly higher answer accuracy than existing methods, particularly for complex questions; (2) high efficiency with minimal LLM calls and token usage; (3) outstanding performance on small-parameter LLMs, highlighting strong practical value.

## 2 RELATED WORK

### 2.1 LLM RESONING WITH PROMPT

Recent advancements in the reasoning capabilities of LLMs have been driven by various prompting techniques aimed at improving their performance on complex tasks. DecomP He et al. (2021) breaks down reasoning tasks into manageable sub-tasks, allowing for step-by-step problem-solving. Chain-of-Thought (CoT) Wei et al. (2022a) and its derivatives, such as Tree-of-Thought(ToT) Yao et al. (2023), Graph-of-Thought(GoT) Besta et al. (2024), and Memory of Thought(MoT) Li & Qiu (2023), have been instrumental in encouraging LLMs to generate intermediate reasoning steps, thereby enhancing their cognitive processes. Plan-and-solve Wang et al. (2024a) prompts LLMs to formulate plans and conduct reasoning based on these plans.

## 2.2 QUESTION ANSWERING WITH KG-AUGMENTD LLM

Knowledge Graph Question Answering (KGQA) has evolved significantly with the integration of LLMs and Knowledge Graphs. Early approaches embedded KG knowledge into LLMs during pre-training or fine-tuning Zhang et al. (2019a), but this limited explainability and update efficiency. To enhance reasoning, retrieval-augmented techniques retrieve relevant facts from KGs Li et al. (2023a). UniKGQA Jiang et al. (2022) unifies graph retrieval and reasoning within a single LLM model, achieving state-of-the-art results. Recent methods translate KG knowledge into textual prompts for LLMs, enhancing reasoning without sacrificing KG's strengths Li et al. (2023b). For instance, generating SPARQL queries or sampling relevant triples aids LLM inference Baek et al. (2023). The KG-augmented LLM paradigm treats LLMs as agents exploring KGs interactively, improving reasoning capabilities Jiang et al. (2023a).

## 3 METHOD

### 3.1 OVERVIEW

The GG-Explore framework operates in two primary phases, each illustrated in its own diagram:

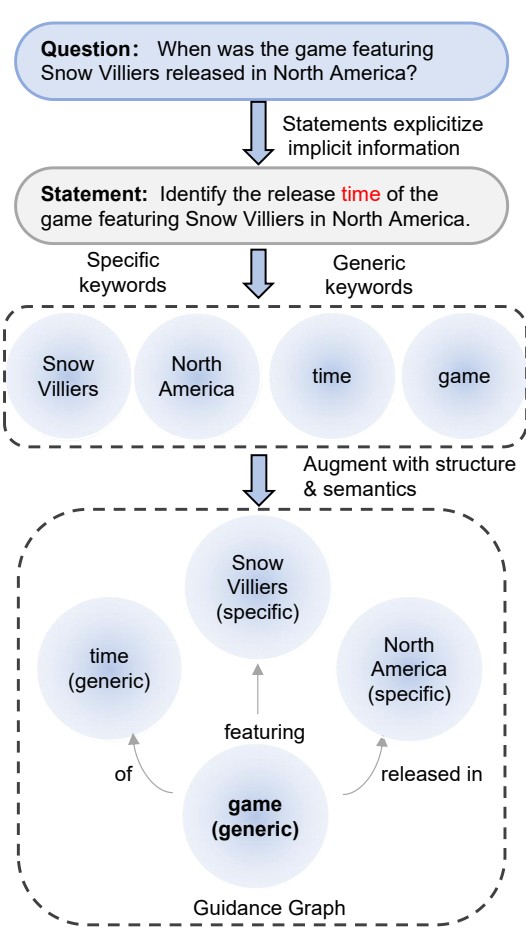

Figure 2: Workflow of Guidance Graph Construction

(1) the construction of a query-specific Guidance Graph (Figure 2), and (2) the iterative exploration of the Knowledge Graph guided by this blueprint (Figure 3).

First, the input question is processed by an LLM to extract fine-grained clues and their semantic relationships, forming the Guidance Graph as detailed in Section 3 and visualized in Figure 2. Then, this graph directs a targeted exploration of the KG through the iterative procedure depicted in Figure 3. Each iteration selects a new target from the Guidance Graph (Section 3.3.1), retrieves candidate entities from the KG, and rigorously filters them through our novel Structural Alignment (Section 3.3.2) and Context-Aware Semantic Alignment (Section 3.3.3) modules. This ensures the retrieved subgraph is both structurally compatible and semantically consistent with the query.

### 3.2 GUIDANCE GRAPH CONSTRUCTION

Our framework constructs Guidance Graphs as illustrated in Figure 2.

We employ LLMs to perform hierarchical processing of questions. First, we transform elliptical questions into complete declarative statements to resolve information incompleteness. Second, we extract fine-grained elements from these statements, categorizing them into: (1) specific keywords (named entities like 'North America') and (2) generic keywords (broader terms like "country"). Third, we mine semantic relationships between keywords to reinforce the structural representation of the Guidance Graph

To establish the graph structure while enriching semantic information, we design the following generation rules based on specific and generic keywords:

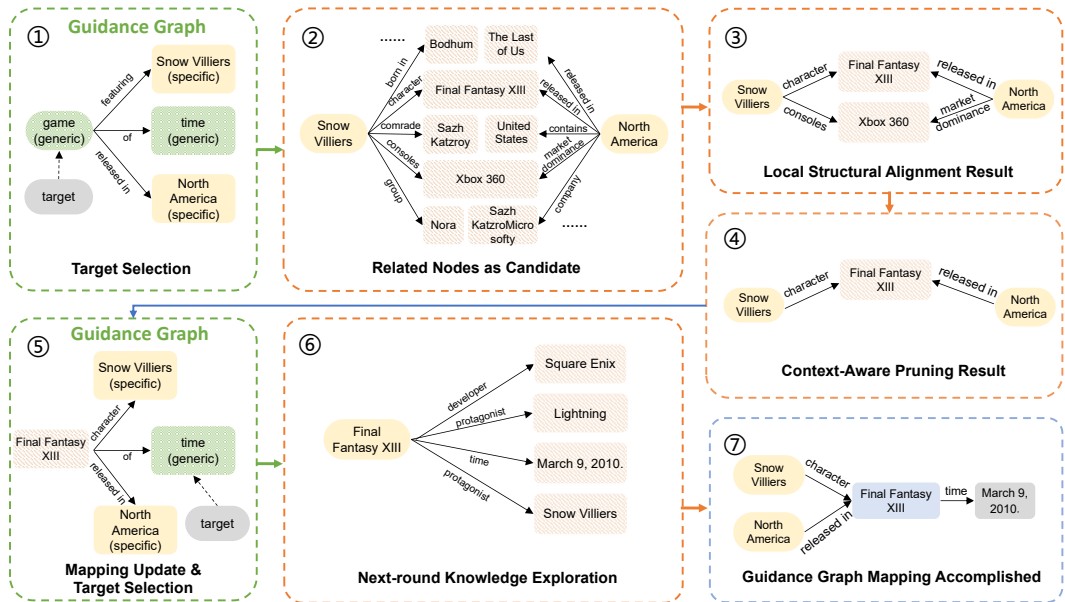

Figure 3: Iterative knowledge exploration maps Guidance Graph nodes and edges to the knowledge graph, finding structurally and semantically matching subgraphs. In step 5, we also performed global consistency alignment, though it was not convenient to display in the figure.

1. Specific keywords exclusively serve as graph nodes (i.e., entities in triples) rather than edges.

2. When a generic keyword co-refers to the same entity as a specific keyword, it functions as a triple's relation rather than an entity.

3. For two distinct generic keywords referencing the same entity, one must be assigned as the relation in the triple.

4. For associated keywords referring to different entities, we construct triples where the keywords become head/tail entities and their association forms the relational edge.

These rules ensure that the structure and semantics of the Guidance Graph closely align with the original intent of the question. To formally describe the forthcoming exploration phase, we denote the set of clue entities (nodes) in $GG$ as $\mathcal{V}_{GG}$. The prompts used are shown in Table A1A2

## 3.3 KNOWLEDGE EXPLORATION

We refer to entities and relations in the Guidance Graph as clue entities and clue relations to distinguish them from their counterparts in the KG. The exploration process (Figure 3) progressively maps these clues to the KG through iterative expansion, ultimately retrieving a subgraph that is structurally and semantically aligned with the Guidance Graph. The prompts used are shown in Table A3A4A5

### 3.3.1 INITIALIZATION AND ITERATIVE TARGET SELECTION

The process begins by identifying *starting points*—specific keywords from the Guidance Graph that correspond to named entities (e.g., "North America"). These are mapped to KG entities via string retrieval, initializing the mapping dictionary:

$$\mathcal{M} = \{\text{clue entity} \mapsto \text{KG entities}\},$$

where $\mathcal{M}(c)$ denotes the set of KG entities currently associated with clue entity $c$. Unlike methods that require all starting points to be identified upfront or rely on impractically broad priors, our approach is robust to partial matches; exploration proceeds as long as at least one starting point is found.

In each iteration, we select the next exploration target—a clue entity $next\_e$—by identifying a triple pattern in the Guidance Graph of the form: $(current\_e, next\_r, next\_e)$ or $(next\_e, next\_r, current\_e)$ where $current\_e$ has already been mapped in $\mathcal{M}$ while $next\_e$ has not. This ensures expansion only occurs from known anchors towards unexplored nodes, preserving the connected structure of the Guidance Graph.

### 3.3.2 STRUCTURAL ALIGNMENT

Conventional pruning methods evaluate entities in isolation, which requires computationally expensive scoring of each candidate against individual relations or entities. To drastically reduce the candidate search space before applying any costly semantic checks, we introduce a **Structural Alignment** module that enforces *global connectivity constraints* from the Guidance Graph in two complementary phases. This model-free mechanism leverages multi-hop structural relationships to efficiently filter a large, noisy candidate set down to a small, structurally coherent subset, providing both presemantic efficiency and post-semantic global coherence.

**Phase 1 – Local Structural Filtering (Pre-semantic).** Let $\Gamma_{\mathrm{KG}}(x)$ be the neighborhood function returning KG entities adjacent to a given KG entity $x$. Given an unexplored target triple

$$(current\_e, next\_r, next\_e),$$

the initial candidate set is:

$$\mathcal{E}_{\mathrm{cand}} = \Gamma_{\mathrm{KG}}(\mathcal{M}(current\_e)),$$

which collects all one-hop neighbors of the current mapped entities in the KG.

To integrate structural evidence from the rest of the Guidance Graph, let $\mathcal{N}_{GG}(next\_e)$ denote the set of clue entities directly connected to $next\_e$ in the Guidance Graph. For each $related\_e \in \mathcal{N}_{GG}(next\_e)$ with a non-empty mapping $\mathcal{M}(related\_e)$, we enforce an auxiliary connectivity constraint: only retain candidates in $\mathcal{E}_{\mathrm{cand}}$ that are also adjacent to at least one entity in $\mathcal{M}(related\_e)$.

Formally, the locally filtered set is obtained by iteratively intersecting candidate neighborhoods:

$$\mathcal{E}'_{\mathrm{cand}} = \bigcap_{\substack{related\_e \in \mathcal{N}_{GG}(next\_e) \\ \mathcal{M}(related\_e) \neq \emptyset}} \{e \in \mathcal{E}_{\mathrm{cand}} \mid e \in \Gamma_{\mathrm{KG}}(\mathcal{M}(related\_e))\}.$$

Each intersection step removes entities that fail to satisfy at least one Guidance Graph connectivity constraint. The outcome is a reduced candidate set $\mathcal{E}'_{\mathrm{cand}}$ that is *structurally coherent* with respect to all mapped anchors seen so far. Crucially, this phase is model-free and lightweight, enabling aggressive pruning before any semantic evaluation, thereby cutting down the search space and the cost of the subsequent alignment stage.

Under KG incompleteness, where structural alignment fails to connect due to missing relations, we let the LLM compare the contextual relevance of alternative branches and select one for continued exploration, permanently pruning the others.

**Phase 2 – Global Consistency Alignment (Post-semantic).** After semantic alignment finalizes the mapping for $next\_e$, we re-apply structural constraints at the *global* level to maintain graph connectivity.

Let $E_{next\_e} = \mathcal{M}(next\_e)$ be the newly added entity set from the current expansion step.

Define $\mathrm{Reach}_{\mathrm{KG}}(e)$ as the set of KG entities reachable from $e$ via any path within the *currently mapped subgraph* induced by $\bigcup_{c \in \mathcal{V}_{GG}} \mathcal{M}(c)$. Reachability can be multi-hop, not restricted to direct neighbors.

We enforce global consistency by retaining only those mapped entities that have a path to at least one of the newly added entities $E_{next\_e}$:

$$\mathcal{M}(c) \leftarrow \{e \in \mathcal{M}(c) \mid E_{next\_e} \cap \mathrm{Reach}_{\mathrm{KG}}(e) \neq \emptyset\}, \quad \forall c \in \mathcal{V}_{GG}.$$

This pruning step removes any previously mapped entities that have lost connectivity with the expanded subgraph. By iteratively maintaining $\mathcal{M}$ as a *single connected component* mirroring the Guidance Graph topology, Phase 2 guarantees that partially valid but globally inconsistent mappings are eliminated, and that subsequent exploration operates on a structurally coherent and connected KG subgraph.

### 3.3.3 CONTEXT-AWARE SEMANTIC ALIGNMENT

The Structural Alignment module produces a structurally-valid but potentially semantically diverse candidate set $\mathcal{E}'_{\text{cand}}$. While these candidates satisfy the global connectivity constraints of the Guidance Graph, the precise semantic meaning of the connection remains unresolved. Existing methods typically address this by performing relation matching based on the entire question or a set of multiple, imprecise fine-grained clues, which often fails to capture the specific contextual meaning of the current exploration step. In contrast, our key insight is that precise semantic disambiguation requires evaluating relations within the minimal, specific narrative context of the current exploration step.

To this end, we design a context-aware semantic alignment that operates on the set of candidate relations $\mathcal{R}$ connecting $\mathcal{E}'_{\text{cand}}$ to $\mathcal{M}(\text{current\_e})$. Note that by working on the relation set $\mathcal{R}$ derived from the already reduced entity set $\mathcal{E}'_{\text{cand}}$, rather than the original large set $\mathcal{E}_{\text{cand}}$, we gain significant computational efficiency, as $\mathcal{R} \ll \mathcal{E}_{\text{cand}}$.

We formulate a minimal yet discriminative context phrase $C$ from the specific Guidance Graph triple (e.g., "current\_e next\_r next\_e"). An LLM is then prompted to select the relation $r \in \mathcal{R}$ that best substitutes next\_r in the phrase $C$ while preserving its meaning. This approach provides two decisive advantages over conventional methods:

1. **Target-awareness:** The pruning is specifically tailored to the current target triple (current\_e, next\_r, next\_e), unlike broad-question-based matching.

2. **Context-awareness:** Semantic matching is performed within this precise, minimal narrative context $C$, preventing ambiguous interpretations that arise from evaluating relations against an entire question or a bag of clues.

The outcome of this step is a semantically precise relation $r$. The final mapping for next\_e is updated to:

$$\mathcal{M}(\text{next\_e}) = \{e \in \mathcal{E}'_{\text{cand}} \mid e \text{ is connected to } \mathcal{M}(\text{current\_e}) \text{ via relation } r\}.$$

The next step is the **Global Consistency Alignment** (Phase 2 of Structural Alignment), which removes any previously mapped entities that lose connectivity in light of this update.

Table 1: Results for WebQSP and CWQ. In CWQ, each question has only one answer, so partial matching is equivalent to complete matching.

| Method | WebQSP | | CWQ | |
|---|---|---|---|---|
| | **partial** | **complete** | **partial** | **complete** |
| *without external knowledge* | | | | |
| IO prompt Brown et al. (2020) w/DeepSeek-V3 | 63.3 | 35.3 | 44.8 | 44.8 |
| COT Wei et al. (2022b) w/DeepSeek-V3 | 70.5 | 41.4 | 46.7 | 46.7 |
| *with external knowledge* | | | | |
| ToG Sun et al. (2024) w/Llama3-8B | 55.6 | 32.3 | – | – |
| PoG Chen et al. (2024) w/Llama3-8B | 63.4 | 34.4 | – | – |
| FiSKE Tao et al. (2025) w/Llama3-8B | 70.8 | 40.4 | – | – |
| StructGPT Jiang et al. (2023b) w/GPT-3.5 | 72.6 | – | 54.3 | 54.3 |
| KB-BINDER Li et al. (2023c) w/GPT-3.5 | 74.4 | – | – | – |
| ToG Sun et al. (2024) w/GPT-3.5 | 76.2 | – | 57.1 | 57.1 |
| PoG Chen et al. (2024) w/DeepSeek-V3 | 81.9 | 60.7 | 55.7 | 55.7 |
| FiSKE Tao et al. (2025) w/DeepSeek-V3 | **82.5** | 61.1 | 50.2 | 50.2 |
| Ours w/Llama3-8B | 79.3 | 54.1 | 56.7 | 56.7 |
| Ours w/DeepSeek-V3 | 81.8 | **64.5** | **71.8** | **71.8** |

## 4 EXPERIMENTS

### 4.1 EXPERIMENTAL SETTINGS

**Datasets and Evaluation Metrics.** To evaluate the performance of our proposed paradigm, We selected one open-source knowledge graph and one self-constructed graph as external knowledge

Table 2: Results for the agricultural graph.

| Method | 1-Hop Query | | Multi-Hop Query | |
|---|---|---|---|---|
| | Partial | Complete | Partial | Complete |
| IO prompt Brown et al. (2020) w/DeepSeek-V3 | 35.8 | 22.1 | 12.4 | 6.6 |
| CoT Wei et al. (2022b) w/DeepSeek-V3 | 37.2 | 23.3 | 12.8 | 7.1 |
| PoG Sun et al. (2024) w/DeepSeek-V3 | 67.7 | 62.6 | 23.9 | 13.0 |
| FiSKE Tao et al. (2025) w/DeepSeek-V3 | 71.1 | 62.8 | 28.3 | 12.9 |
| Ours w/DeepSeek-V3 | **89.9** | **75.8** | **81.8** | **56.8** |

Table 3: Efficiency Comparison Across Datasets. The agr denotes the agricultural dataset.

| Dataset | Method | LLM Call | Input Token | Output Token | Total Token |
|---|---|---|---|---|---|
| WebQSP | PoG | 11.3 | 6590.2 | 427.0 | 7017.2 |
| | FiSKE | 7.9 | 3079.0 | 1379.7 | 4458.7 |
| | **Ours** | 8.6 | 3264.5 | 584.0 | 3848.5 |
| CWQ | PoG | 23.4 | 15483.4 | 694.8 | 16178.2 |
| | FiSKE | 9.4 | 3578.8 | 1828.7 | 5407.5 |
| | **Ours** | 10.2 | 4052.0 | 708.6 | 4760.6 |
| agr-one-hop | PoG | 5.6 | 2233.9 | 217.4 | 2451.4 |
| | FiSKE | 5.9 | - | - | - |
| | **Ours** | 4.6 | 1000.0 | 294.4 | 1294.4 |
| agr-multi-hop | PoG | 6.9 | 3243.6 | 379.6 | 3623.3 |
| | FiSKE | 13.1 | - | - | - |
| | **Ours** | 4.1 | 975.5 | 380.7 | 1356.2 |

bases for experimentation. The open-source knowledge graph is Freebase Bollacker et al. (2008). The graph that we constructed ourselves will be publicly available soon and is referred to in this paper as the agricultural knowledge graph.

Freebase is a large-scale, semi-structured database supported by Google, designed to collect and connect information about millions of entities and their relationships worldwide. With its exceptionally large volume of relations and entities, Freebase perfectly fits our knowledge-dense scenario. We conduct experiments on two QA sets with Freebase as external knowledge base: WebQSP Yih et al. (2016) and CWQ Talmor & Berant (2018).

The agricultural knowledge graph that we constructed includes over 100,000 entities and 1 million triples. Although the number of entities is relatively modest compared to general-purpose knowledge graphs like Freebase, its relational structure remains highly complex (as shown in Appendix A.3). On this basis, we built single-hop and multi-hop question-answering sets.

We evaluate multi-answer questions through two distinct metrics: (1) partial match, where success requires retrieval of at least one correct answer, and (2) complete match, which necessitates identification of all correct answers.

**Baselines.** We compared our approach with seven baseline methods: standard prompting (IO prompt) Brown et al. (2020), chain of thought prompting (CoT prompt) Wei et al. (2022b), ToG Sun et al. (2024), PoG Chen et al. (2024), StructGPT Jiang et al. (2023b), KB-BINDER Li et al. (2023c) and FiSKE Tao et al. (2025). IO prompt and CoT prompt are two knowledge-free methods, used to measure how many questions LLMs can answer solely based on their internal knowledge.

Table 4: Experimental results regarding starting points. In Filtered-CWQ, questions without identifiable starting points were removed. In our approach, such questions are directly answered by the LLM.

| Dataset | Method | LLM Call | Input Token | Output Token | Total Token | Partial match | Complete match |
|---|---|---|---|---|---|---|---|
| CWQ | **Ours** | 10.2 | 4052.0 | 708.6 | 4760.6 | 0.7179 | 0.7179 |
| Filtered-CWQ | **Ours** | 12.0 | 4875.4 | 769.9 | 5645.3 | 0.7460 | 0.7460 |

Table 5: Ablation Studies on CWQ set.

| Variant | partial match | LLM Call | Input Token | Output Token | Total Token |
|---|---|---|---|---|---|
| Ours | 71.8 | 10.2 | 4052.0 | 708.6 | 4760.6 |
| w/o Context-Aware Semantic Alignment | 53.8 | 7.9 | 2908.5 | 591.1 | 3499.6 |
| w/o Structural Alignment | 67.9 | 11.6 | 4761.9 | 829.5 | 5591.4 |
| w/o Dynamic Branch Selection | 69.2 | 10.0 | 3983.8 | 701.4 | 4685.2 |

ToG, PoG, StructGPT and KB-BINDER represent previous state-of-the-art approaches in knowledge base question answering.

**Experiment Details.** To ensure the reliability and reproducibility of the experiments, we set the temperature parameter to 0 for all LLMs. For the English QA datasets (WebQSP and CWQ), we employed the original prompts provided in the baseline method's codebase. For the Chinese agricultural QA dataset, we adapted Chinese versions of the prompts by translating the English template while preserving the structural format for each baseline method.

## 4.2 PERFORMANCE COMPARISON

The results of WebQSP and CWQ are shown in Table 1. On the WebQSP dataset, our approach with Llama3-8B-Instruct significantly outperforms existing methods using the same model, achieving results comparable to or even surpassing those obtained by large-scale models like GPT-3.5 or DeepSeek-V3 DeepSeek-AI et al. (2024). This not only demonstrates the effectiveness of our design but also highlights its particular suitability for smaller LLMs, indicating substantial practical value. When evaluated on DeepSeek-V3, our method achieves partial match performance comparable to the state-of-the-art FiSKE and PoG methods while outperforming all other approaches, and exceeds all methods in complete match results. This suggests our approach tends to retrieve more comprehensive knowledge.

Compared to WebQSP, the CWQ dataset contains more multi-hop questions. On CWQ, FiSKE underperforms PoG, validating our claim that relying solely on fine-grained information struggles with complex problems. Our method improves upon FiSKE over 20 percentage points, demonstrating its strong capability in knowledge-intensive scenarios for solving complex problems.

The experimental results on the agricultural dataset are presented in the Table 2. For fair comparison, we equipped PoG with the same fine-grained information extraction method as ours, since its original implementation relies on dataset-provided topic entities which are unavailable in agricultural QA sets. Our method demonstrates significant improvements over existing approaches, particularly for multi-hop question answering, where the advantage reaches remarkable levels. A notable observation reveals that while existing methods show substantial performance gaps between single-hop and multi-hop QA tasks, our approach consistently maintains high performance across both scenarios. This demonstrates our method's superior capability in handling complex problems within knowledge-intensive graphs, as also evidenced in Table 1.

Overall, our method demonstrates superior performance compared to existing approaches, particularly excelling in knowledge-intensive scenarios for solving complex problems while maintaining strong compatibility with smaller-scale LLMs.

## 4.3 COMPUTATIONAL COST

We conducted an efficiency study of our method, with results shown in the Table 3. Overall, our analysis reveals that our approach demonstrates comprehensive advantages over PoG in both the number of LLM calls and total token consumption. On WebQSP (dominated by single-hop questions), our method shows modest improvements, while on CWQ (containing primarily multi-hop questions), it achieves significant gains, highlighting its superior capability in handling complex problems.

On the agricultural dataset, our method requires slightly fewer LLM calls than PoG, while the token consumption by the LLM is significantly reduced. We attribute this to PoG's reliance on heavily constrained prompts with strict rules to govern LLM outputs, whereas our approach embeds constraints primarily within the designed strategies rather than the prompts. This distinction may explain why

our method is more friendly to smaller LLMs. FiSKE demonstrates high efficiency on WebQSP and CWQ, but incurs significant computational overhead when handling the structurally complex agricultural dataset. In contrast, our approach consistently achieves stable and superior performance across all datasets.

### 4.4 STUDIES ON STARTING POINTS

We conducted experiments to evaluate the impact of starting point retrieval success on our proposed method with results presented in Table 4. When the starting point retrieval fails, our method defaults to having the LLM answer the question directly. It should be noted that question-guided methods like ToG and PoG directly retrieve topic entities as starting points from the dataset.

The results demonstrate that bypassing questions without identifiable starting points leads to improved answer accuracy at a marginal cost to efficiency. This observation is well-justified: while our method achieves significantly higher accuracy than direct LLM responses, it naturally incurs additional computational overhead.

Overall, while the inability to identify starting points affects our method's performance, this limitation does not outweigh its substantial advantages.

### 4.5 ABLATION STUDIES

We conducted ablation studies and the results are shown in Table 5. The impact of starting points was discussed in the previous subsection. We therefore evaluate the effect of Context-Aware Semantic Alignment and Structural Alignment — including its integrated dynamic branch selection mechanism for handling KG incompleteness.

For Context-Aware Semantic Alignment, we replace the original semantically rich target phrases with single clues. For Structural Alignment, we test two settings: (1) removing the entire Structural Alignment module, and (2) keeping Structural Alignment but disabling the dynamic branch selection mechanism used when relations are missing.

The ablation results demonstrate the necessity of each component. Replacing semantic context with single clues in Context-Aware Semantic Alignment causes significant performance drops, highlighting the crucial role of precise contextual semantics in LLM-based pruning. Removing Structural Alignment reduces both accuracy and efficiency, confirming its dual benefits as designed. Disabling the dynamic branch selection mechanism slightly reduces computational overhead but at the cost of performance, justifying its trade-off of minimal token consumption for substantial accuracy gains.

Another key characteristic of our method is its sequential nature, specifically manifested in the target selection and expansion process. Both Context-Aware Semantic Alignment and Structural Alignment (including dynamic branch selection) function most effectively when executed in this ordered framework, and their collective impact reinforces the necessity of maintaining this sequential pipeline.

## 5 CONCLUSION

In this paper, we propose Guidance-Graph-guided Knowledge Exploration, a novel paradigm that bridges the gap between unstructured queries and structured knowledge retrieval. By introducing the Guidance Graph, our method resolves the dilemma faced by existing approaches that either suffer from redundant exploration or struggle to distinguish complex structures. Our method extracts multi-dimensional information from questions and establishes systematic construction rules for Guidance Graph, endowing it with both rich structural and semantic information. We propose Structural Alignment and Context-Aware Semantic Alignment to efficiently and accurately identify knowledge subgraphs that maintain both structural and contextual semantic consistency with the Guidance Graph. Extensive experiments demonstrate that our method achieves significant improvements across multiple datasets, particularly demonstrating exceptional capability in distinguishing complex structures within knowledge-intensive scenarios. The results further reveal our method's strong compatibility with smaller-scale LLMs, highlighting its practical value. Comprehensive ablation studies validate the contribution of each proposed module.

**Reproducibility statement** We have submitted our core code, which includes the complete algorithm logic along with comprehensive annotations. The referenced utils.py file contains functions for large language model invocations and database access, none of which impact the core algorithm. The prompts used for the LLM are provided in the appendix, and the return values of the database access functions are explicitly annotated within the main code. Based on the submitted code and the prompts in the appendix, our proposed method can be easily reproduced.

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

# A   APPENDIX

## A.1   LLM USAGE STATEMENT

The large language model was used solely for grammar checks and polishing, and no other purposes.

## A.2   PROMPT TEMPLATES

The prompts for Guidance Graph construction includes 'information extraction' A1 and 'GG construction' A2. The prompts for Knowledge Exploration includes 'relation pruning' A3, 'choose' A4 and 'answering' A5. The prompt 'choose' is for dynamic branch selection. When you need to switch between different datasets, you only need to modify the examples in the prompts.

Table A1: information extraction.

| | |
|---|---|
| information extraction | Rephrase the question as a statement, ensuring:
1. \*\*All explicit and logically implied information is included\*\* (e.g., location, time, scope if hinted in the question).
2. Split compound noun phrases into individual entities.
3. Classify each entity as generic/specific.
Do not answer the question.
question: What team did Payton Manning's father play for?
answer:
statement: Identify the team payton manning's father played for.
entities:
- team (generic)
- payton manning (specific)
- father (generic)
question: what did james k polk do before he was president?
answer:
statement: Identify the job of James K. Polk before he became president.
entities:
- job (generic)
- James K. Polk (specific)
- president (generic)
question: Where did the "Country Nation World Tour" concert artist go to college?
answer:
statement: Identify the college attended by the concert artist of the "Country Nation World Tour."
entities:
- college (generic)
- Country Nation World Tour (specific)
- concert artist (generic)
Now answer with the format of the example above. Be brief and precise.
question: {}
answer: |

Table A2: GG construction.

| | |
|---|---|
| GG construction | Please analyze the relationships between the following entities in the given sentence.
Represent each relationship as a triple in the format (subject, relation, object).
These keywords can be used as entities or relationships in the triples, but please do not modify the keywords.
If a generic keyword and another keyword refer to the same entity, the generic keyword should be treated as a relation rather than an entity.
sentence: Identify the father of Keyshia Cole.
generic keywords: father
specific keywords: Keyshia Cole
answer:
triples: [(father, of, Keyshia Cole)]
sentence: Identify the main trading partner of China that appointed Abdelaziz Bouteflika to a governmental position.
generic keywords: main trading partner; governmental position
specific keywords: China; Abdelaziz Bouteflika
answer:
triples: [(main trading partner, of, China), (main trading partner, appointed to governmental position, Abdelaziz Bouteflika)]
Now answer with the format of the example above. Be brief and precise.
sentence: {}
generic keywords: {}
specific keywords: {}
answer: |

## A.3   COMPLEX STRUCTURE OF AGRICULTURAL KNOWLEDGE GRAPHS

In agricultural knowledge graphs, diverse relationships may exist between different nodes, creating multiple possible pathways between the same nodes. In contrast, general knowledge graphs often exhibit a relatively simpler structure, where there might be only one pathway between different nodes.

Table A3: relation pruning

| relation pruning | Select the most semantically matching relation from the candidate relations to replace the given relation in the theme information while maintaining its essential meaning. Rate it on a scale of 0 to 10. Be brief and precise. Rate it on a scale of 0 to 10. Be brief and precise. Below is an example: theme information: main spoken language of country relation: of candidate relations: ['language.human_language.main_country', 'language.human_language.human_language', 'language.human_language.language_family', 'language.human_language.iso_639_3_code', 'base.rosetta.languoid.parent', 'language.human_language.writing_system', 'base.rosetta.languoid.languoid_class'] assess: - 'language.human_language.human_language' can perfectly replace the relation, (9) score. - 'language.human_language.main_country' can replace the relation, (8) score. - other relations are not related to any information in the sentence, (0) score. Now answer with the format of the example above. Be brief and precise. theme information:{} relation: {} candidate relations: {} assess: |
|---|---|

Table A4: choose.

| choose | Based on the target sentence below, determine which of the provided pieces of knowledge better matches the target sentence. Be breif and precise. Target sentence: Identify the country that Turkey trades with which contains the Annaba Province. Knowledge 1: country location.adjoining_relationship.adjoins turkey Knowledge 2: country location.contain.contains Annaba Answer: Knowledge (2) better matches the target sentence. Target sentence: {} Knowledge 1: {} Knowledge 2: {} Answer: |
|---|---|

Table A5: answering.

| answering | Please answer the question. Triples are available for reference. If there are not enough information in the triples, please answer with your own knowledge. question: Which place is the madam satan located? triplets: [('madam satan', 'film.film.country', 'the USA'), ('madam satan', 'film.film.language', 'English')] answer: 'madam satan' is located in 'the USA'. question: {} triplets: {} answer: |
|---|---|

As shown in the Fig. 4, four nodes in an agricultural knowledge graph are connected by five types of relationships, whereas only three types of relationships link four nodes in a general knowledge graph. This difference in relationship variety leads to varying levels of structural complexity.

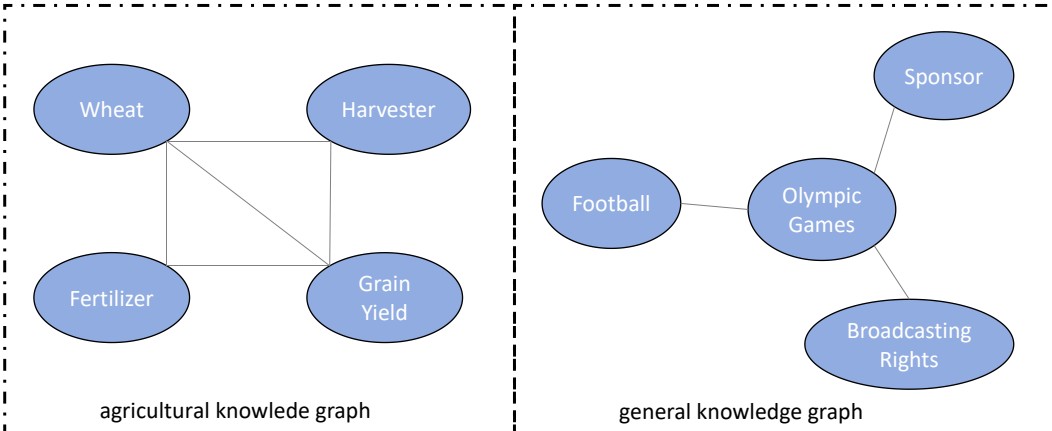

Figure 4: This difference in relationship variety leads to varying levels of structural complexity between agricultural knowledge graph and general knowledge graph.

