# OpenReview forum: "Guided Navigation in Knowledge-Dense Environments: Structured Semantic Exploration with Guidance Graphs"
_ICLR.cc/2026/Conference — ICLR 2026 Conference Withdrawn Submission_

### Official Review · Reviewer_3mVo · 2025-10-15

**Soundness:** 2
**Presentation:** 2
**Contribution:** 2
**Rating:** 4
**Confidence:** 4

**Summary:**

This paper addresses the challenge of granularity mismatch in KG-based question answering, where natural language queries provide high-level semantic cues while KG exploration requires precise structural navigation. The authors propose GG-Explore, a two-stage framework that first constructs a Guidance Graph — a lightweight, query-specific semantic graph that extracts specific and generic keywords and encodes them as a structured reasoning blueprint — and then performs guided KG traversal via a combination of structural alignment (fast pruning based on graph constraints) and context-aware semantic alignment using LLM judgment. This approach explicitly bridges semantic intent and KG structure, reducing redundant exploration and improving multi-hop reasoning efficiency on WebQSP, CWQ, and a large agricultural KG, outperforming baselines such as PoG and FiSKE, and enabling small LLMs to match or surpass GPT-3.5 in complex KG navigation tasks.

**Strengths:**

1. Clear problem formulation — The paper clearly identifies the gap between semantic query interpretation and structural KG traversal and motivates the need for an intermediate reasoning representation rather than direct retrieval.
2. Guidance Graph as a structured semantic scaffold — Introducing a query-specific semantic graph as a blueprint before KG exploration is a neat idea that balances flexibility (LLM reasoning) and precision (structured navigation), improving interpretability.
3. Effective two-stage alignment mechanism — The combination of structural pruning followed by context-aware semantic filtering provides a principled way to reduce KG search space while preserving answer coverage, rather than relying on purely semantic matching.
4. Lightweight yet effective implementation — The method does not require large LLMs or heavy graph neural modules, yet achieves notable improvements, showing that guidance and structure can outperform raw model scale.
5. Strong empirical results on both standard KG QA benchmarks and a large real-world agricultural KG, with particularly strong gains on multi-hop tasks, demonstrating robustness beyond toy settings.

**Weaknesses:**

1. Novelty is somewhat incremental. While the paper positions Guidance Graph as a new semantic-structural bridge, many prior works utilize intermediate graph-like cues for KG traversal. The distinction between those “cue graphs” and the proposed Guidance Graph is not sharply articulated, making the conceptual contribution appear more like a refined engineering pipeline than a fundamentally new formulation.
2. Pipeline complexity is relatively high. The full system involves keyword typing, semantic blueprint construction, structural pruning, semantic filtering, and multi-step KG search, which makes the approach heavyweight compared to more elegant end-to-end retriever–LLM paradigms. The paper does not show whether all components are necessary or whether a simpler guidance signal could achieve similar gains.
3. Generalization beyond curated KG datasets is unclear. The method requires structured KG traversal under strict graph constraints, but it is not evaluated on open-domain, incomplete, or noisy graphs, where structural guidance may be brittle or misaligned with real graph topology.
4. Efficiency vs. complexity trade-off is under-discussed. The structured pruning may reduce KG expansion cost, but the overhead of constructing Guidance Graph + LLM semantic alignment steps may offset the gains, especially in large-scale industrial KG settings. No clear computational cost breakdown is provided.
5. Limited discussion of failure modes / robustness. If the Guidance Graph extracted by LLM is partially wrong or incomplete, it is unclear whether the system recovers or gets stuck due to hard structural filtering—a potential failure mode not analyzed in the experiments.

**Questions:**

See weaknesses.

---

### Official Review · Reviewer_vvsd · 2025-10-29

**Soundness:** 2
**Presentation:** 1
**Contribution:** 2
**Rating:** 2
**Confidence:** 4

**Summary:**

An LLM-based KGQA method is proposed. An intermediate graph representation is converted from the query by rule-based keyword mapping. Then it is grounded to the KG with the help of LLM for relation disambiguation and using structural connectivity.

**Strengths:**

S1. Some code is provided.

S2. Computational cost is reported.

**Weaknesses:**

W1. The presentation is hard to follow.
- Figure 3 is confusing: terms differ from those in the text; rounds are not clearly separated; the "target" node is misleading.
- The running example cannot cover all the main steps of the approach.
- The input and output of each step is not clear.
- One has to jump from Phase 1 in Section 3.3.2 to 3.3.3 and then back to Phase 2 in Section 3.3.2.

W2. Evaluation is insufficient.
- Only two public datasets are used. Considering the heuristic nature of the proposed approach which heavily relies on the effectiveness of rules, more datasets such as GrailQA should be included.
- The baselines are a bit old, all published in or before 2024 except for a journal paper published in 2025. How about more recent baselines such as RoG, GNN-RAG, EPERM, FiDeLiS, Paths-over-Graph, Debate-on-Graph, FastToG, ReKnoS, GCR, REKG-MCTS?
- The ablation study should separately test the effectiveness of Phase 1 and Phase 2 structural alignment.
- A closed-source dataset is used. It is unclear whether it will be open-source. The results are currently not reproducible.

W3. Novelty is unclear---seems limited.
- Comparison with related work is short and lacks comparative analysis.
- The proposed approach heavily relies on rules (Section 3.2) and simple structural connectivity (Section 3.3.1).

**Questions:**

Q1. Could you formulate the input and output of each step, illustrated using your running example?

Q2. Why using a closed-source dataset instead of other popular public datasets such as GrailQA?

Q3. Could you compare your results with more recent baselines?

Q4. What is the technical novelty of your approach?

---

### Official Review · Reviewer_nsHd · 2025-11-01

**Soundness:** 2
**Presentation:** 2
**Contribution:** 2
**Rating:** 6
**Confidence:** 3

**Summary:**

This paper proposes GG-Explore, a novel framework addressing the limitations of LLMs in knowledge-intensive tasks by integrating them with KGs through an intermediate "Guidance Graph." This graph acts as a semantic blueprint, constraining the search space for knowledge retrieval using a hybrid pruning strategy. GG-Explore achieves superior efficiency and outperforms state-of-the-art methods, even with smaller LLMs, especially on complex tasks. However, the paper exhibits several shortcomings, including insufficient experimentation, formatting issues, and unclear diagrammatic representations. Specific critiques are detailed below:

**Strengths:**

Please see stregthness

**Weaknesses:**

1. The ablation experiments are notably incomplete, being conducted on a single dataset and solely focusing on "Efficiency Comparison" rather than a comprehensive analysis of individual component contributions to overall performance.

2. While claiming to be state-of-the-art, the experimental results do not consistently demonstrate superior performance.

3. The paper contains several stylistic and formatting errors, such as missing periods on lines 42, 159, and 198, which detract from its professional presentation.

4. The distinctions between the various sub-components presented in Figure 2 are not clearly elucidated. The authors should provide more precise labels and explanations within the figure itself to enhance clarity.

**Questions:**

Please see weakness

---

### Official Review · Reviewer_Mcqv · 2025-11-04

**Soundness:** 2
**Presentation:** 2
**Contribution:** 2
**Rating:** 4
**Confidence:** 4

**Summary:**

A knowledge exploration framework, GG-Explore, is proposed that exploits a query-time-maintained guidance graph to summarize structural and contextual constraints; two search mechanisms, for structural alignment and context-aware semantic alignment, are introduced to leverage the guidance graph to improve information retrieval in KGs. The former

**Strengths:**

S1. A guided graph model may help retrieve more relevant results more quickly in large KGs.
S2. The computational process has been described with sufficient detail.
S3. An experimental study uses a blend of small-scale LLMs to verify the claimed results.

**Weaknesses:**

W1. It lacks the clarity of an overall computational problem with a quantifiable design goal.
W2. There is little formal justification for the strategies (as is, it seems the main strategy relies on a set of heuristics) - more needs to be shown in terms of potential guarantees of relevance, correctness, or efficiency.
W3. There seems to be little discussion of LLM optimization, consistency guarantees, and uncertainty, etc.
W4. The presentation needs significant improvement.

**Questions:**

D1. There is a trade-off between the additional overhead of constructing query-specific guidance graphs and the improved query accuracy they enable. For example, no formal analysis of time cost is provided—this may make it hard to evaluate how the methods scale to a large set of query workloads and whether they scale well when deployed to real data systems.

D2. The overall process is described with a sequence of heuristic strategies. It is not very clear how specific technical challenges are highlighted and addressed, and how these strategies are intuitively linked to improved results. On the other hand, KGQA and graph RAG systems with knowledge retrieval or query templates have been studied, and more baselines from there need to be discussed.

Abujabal, Abdalghani, et al. "Automated template generation for question answering over knowledge graphs." Proceedings of the 26th International Conference on World Wide Web. 2017.

D3. The evaluation also needs to be enriched with a broader range of KGs. Freebase, as a Web knowledge base, already has well-equipped ontologies, making the work less compelling. It is suggested that the authors consider more domain-specific KGs where schema or ontologies are hard to obtain, thereby providing a better motivation scenario.

D4. There seems to be no particular discussion on how to reduce LLM calls and prompt size.

D5. Consider providing an algorithmic description of the major steps to provide a clearer view of the components and the overall automated dataflow -- especially how the system handles a large query workload, a mix of queries with different difficulty levels, and other scalability concerns.

---

### Author Response · Authors · 2025-12-02
**General Response**

Dear Reviewers,

We sincerely thank all of you for your time and thoughtful reviews of our submission. Your constructive feedback has been invaluable in identifying areas for improvement, and we particularly appreciate the comments regarding the insufficiency of theoretical analysis in our work.

We would like to clarify that our method is indeed motivated by theoretical insights, and we have strived to innovate based on these foundations. However, upon reflection, we acknowledge that the current version of the paper does not fully articulate the theoretical basis and rigor underpinning our approach, which may have led to misunderstandings about its strengths and novelty.

Due to time constraints during the rebuttal period, we find ourselves unable to revise the manuscript in a way that fully addresses these concerns. As such, we have decided to withdraw the submission in order to make significant improvements, including expanding the theoretical analysis and presenting our approach in a more comprehensive manner.

We deeply appreciate your constructive feedback, which has helped us identify areas for improvement and provided thoughtful guidance for us to refine our work.

---

### Note · Authors · 2025-12-02

I have read and agree with the venue's withdrawal policy on behalf of myself and my co-authors.